# Different Representation of Mesoscale Convective Systems in Convection-Permitting and Convection-Parameterizing NWP Models and Its Implications for Large-Scale Forecast Evolution

**Karsten Peters** [1,*,†] , **Cathy Hohenegger** [1] and **Daniel Klocke** [2]

1   Max-Planck-Institut für Meteorologie, Bundesstraße 53, 20146 Hamburg, Germany
2   Deutscher Wetterdienst, Frankfurter Straße 135, 63067 Offenbach am Main, Germany
*   Correspondence: peters@dkrz.de
†   Current address: Deutsches Klimarechenzentrum GmbH (DKRZ), Bundesstraße 45a, 20146 Hamburg, Germany.

**Abstract:** Representing mesoscale convective systems (MCSs) and their multi-scale interaction with the large-scale atmospheric dynamics is still a major challenge in state-of-the-art global numerical weather prediction (NWP) models. This results in potentially defective forecasts of synoptic-scale dynamics in regions of high MCS activity. Here, we quantify this error by comparing simulations performed with a very large-domain, convection-permitting NWP model to two operational global NWP models relying on parameterized convection. We use one month's worth of daily forecasts over Western Africa and focus on land regions only. The convection-permitting model matches remarkably well the statistics of westward-propagating MCSs compared to observations, while the convection-parameterizing NWP models misrepresent them. The difference in the representation of MCSs in the different models leads to measurably different synoptic-scale forecast evolution as visible in the wind fields at both 850 and 650 hPa, resulting in forecast differences compared to the operational global NWP models. This is quantified by computing the correlation between the differences and the number of MCSs: the larger the number of MCSs, the larger the difference. This fits the expectation from theory on MCS–mean flow interaction. Here, we show that this effect is strong enough to affect daily limited-area forecasts on very large domains.

**Keywords:** convection-permitting model; mesoscale convective systems; convection parameterization; high-resolution NWP; convection–large-scale interaction; West African squall lines

## 1. Introduction

What is it in numerical atmospheric models that puts the largest constraints on our ability to provide more trustworthy simulations of weather and climate? When asked this question, many model deficiencies would come to mind, most of them related to the conceptual models used to approximate the effect of physical processes unresolved by the atmospheric models' grid spacing on the evolution of the resolved dynamical and thermodynamical model fields. Of these conceptual models, or parameterizations, the ones applied to represent atmospheric moist convection and the associated vertical redistribution of momentum, heat, and moisture are seen as the main contributors to the uncertainty in projections of future climate (see [1] and the references therein). Although decades of research have gone into the understanding of atmospheric convection and the design of parameterizations, reality has caught up with us in recent years and has revealed the dreary state we find ourselves in (e.g., [2]). As a result, the atmospheric science community is currently in the

process of bundling resources to further advance our understanding of atmospheric convection and its representation in atmospheric models [3–7].

However, investing resources into the development of a new generation of convection parameterizations for use in general circulation models is not the only avenue to take. The ever-increasing availability of computational resources has sparked serious interest in attenuating the "cumulus parameterization problem" (coined so by [8]) by focusing efforts on exploring effective ways of running atmospheric models at convection-permitting grid spacings on large domains, even global, and long time periods (e.g., [9–16]). Such models use grid spacings of a few kilometers, allowing them to resolve convective storms explicitly.

The most obvious and by now well-documented improvements obtained when switching off the convection parameterization pertain to a better representation of surface precipitation statistics, such as the timing of the diurnal cycle or the distribution of precipitation intensity (e.g., [15,17–19]). Recent studies have shown that the characteristics of mesoscale convective systems (MCSs) can be adequately represented in climate-scale simulations run with convection-permitting grid spacing [20], and that convection organizes more realistically in simulations with explicit treatment of convection [21]. Despite their shortcomings, well-tuned NWP models with convection parameterization are able to capture the precipitation climatology remarkably well (e.g., [22,23]). This motivates the following question:

> Does the representation of MCSs in very large-domain, convection-permitting NWP models show a measurable effect on the large-scale forecast evolution compared to state-of-the-art global NWP models using parameterized convection?

This question is particularly interesting as it is generally assumed that, in limited-area NWP model simulations, the large-scale circulation is fully constrained by the boundary conditions. The latter are set by the parent model, which, for the time being and in operational applications, has to rely on a convective parameterization.

We attempt to shed light on the above-mentioned matter by analyzing a set of three NWP-model simulations over Western Africa covering the time period 1–31 August 2016. Two models represent current state-of-the-art operational global NWP models with grid spacing $\mathcal{O}$ (10 km) and convection parameterization, and the third model was run at a convection-permitting horizontal grid spacing of 2.5 km over a very large domain covering the entire tropical Atlantic (cf. Section 2.1; northwestern and southeastern corners at 20° N, 68° W and 10° S, 15° E, respectively, [12,16]). We used daily forecasts from all three models. Of course, effects of MCSs on a large scale would be more prominent in a free-running forecast spanning a longer time period. In practice, however, weather forecast centers will not perform free-running convection-permitting simulations, but restart them daily. In this sense, our approach displays a more stringent test to obtain an idea of the actual impact on the large-scale circulation in forecast mode.

Compared to our study, previous studies applying convection-permitting models over that region mainly focused on the ameliorated representation of MCSs and precipitation statistics (e.g., [18,22–25]). Further, the setup we use here covers a substantially larger domain and longer time period for analysis. The very large domain allows for a relatively unconstrained evolution of the synoptic scale conditions, whereas the long analysis period allows sampling various synoptic regimes and quantifying the representation of the climatology of the region. Both aspects may be significantly influenced by the representation of MCSs in models, making the convection-permitting and convection-parameterizing simulations we use here the ideal basis to approach the above-framed question.

Current literature lacks studies on the specific evaluation of the impact of a biased representation of MCSs in convection-parameterizing models on large-scale wind fields on daily timescales under realistic conditions. Previous work however suggested that this effect could be large and significant. A 24-h simulation, [26] showed that Sahelian squall lines influence the mean wind fields up to 700 km away from the system itself; the exact mechanisms remain unclear, though the work

in [27] found a temporary disruption of the African Easterly Jet in the wake of long-lived MCSs in convection-permitting simulations. Observations indeed suggest that understanding and representing the interaction of MCSs with the large-scale circulation is paramount for comprehending both weather on daily timescales, as well as the climatology of the region [28]. The work in [29] found a weaker West African Monsoon circulation when convection was explicitly resolved instead of being parameterized in NWP models. Here, we analyze the impact of a biased representation of MCSs on wind fields at 850 and 650 hPa on daily timescales in convection-parameterizing models, therefore expanding on the findings of existing literature.

In our analysis, we focus on the representation of westward-propagating mesoscale precipitation features, which take the form of MCSs (squall lines) in the convection-permitting simulations, and their effect on the evolution of the forecasted large-scale circulation in all three models, which we introduce in Section 2. We analyze the general characteristics of westward-propagating mesoscale precipitation features in Section 3.1 and quantify differences in forecast evolution between the convection-permitting and convection-parameterizing models throughout the troposphere within a 24-h forecast lead time in Section 3.2. We link these differences to the representation of MCSs in the convection-permitting model and the corresponding absence of propagating MCSs in the models relying on convective parameterization (Section 4). Section 5 summarizes our findings and concludes this paper.

## 2. Materials and Methods

### 2.1. Simulations and Observations

We made use of daily ICON (Icosahedral Non-hydrostatic model [30]) forecasts over the tropical Atlantic region performed at convection-permitting resolution (2.5 km average square root of the grid cell area [12]), denoted as ICON-HiRes in the following. The ICON-HiRes forecasts were initialized daily at 00 UTC from the European Centre for Medium-Range Weather Forecasts (ECMWF) analyses and then run for 36 h. Lateral boundary conditions were provided by the corresponding ECMWF operational forecast made with the Integrated Forecast System (IFS) every three hours. The first 12 h of the simulations were considered as model spin-up and disregarded for analysis. ICON-HiRes was set up identically to the operational global NWP setup at Deutscher Wetterdienst (DWD), ICON-oper, with the exception that the convection (shallow and deep), gravity-wave drag, and sub-grid scale orography parameterizations were turned off. Furthermore, the microphysics parameterization also included graupel as the prognostic variable. The model top was at 30 km, and 75 vertical levels were used. More details on the simulation setup and an evaluation of some aspects of the simulations can be found in [12,31]. An evaluation of bulk atmospheric quantities, which were represented surprisingly well in ICON-HiRes despite being an untuned model version, as well as specific aspects like the diurnal cycle are given in [16].

The two operational forecast products we used were performed with the forecast systems of the German Weather Service (DWD) and of the ECMWF. The two products are denoted as ICON-oper and IFS, respectively. Both global forecast systems parameterized convection and were run with 13-km (ICON-oper) and 9-km (IFS) nominal grid spacing. For ICON-oper, the model top was at 75 km, and 90 vertical levels were used (see [30] for further details). For IFS, the model top was at 0.01 hPa, and 137 vertical levels were used (see [32] for further details). For ICON-oper and IFS, we used daily forecasts initialized at 00 UTC. Both models were initialized from their own analysis resulting from their own data assimilation procedures. For reasons of consistency with the ICON-HiRes data used for analysis, we also only analyzed data covering forecast lead times of 13–36 h. While ICON-oper and IFS differed in the majority of the employed parameterization schemes, they shared the same convection scheme [33,34], although it should be noted that the convection scheme was tuned differently between the two models.

The analysis period was 1–31 August 2016, i.e., the time period of the Next-Generation Aircraft Remote Sensing for Validation II (NARVAL-II) field campaign, which took place in the tropical western Atlantic [6].

Observational estimates of total precipitation were taken from the TRMM Multi-satellite Precipitation Analysis (TMPA [35,36]).

### 2.2. Analysis Methodology

To investigate the effects of MCSs on the large-scale circulation, we needed to isolate them. We identified westward-propagating mesoscale precipitation features and quantified their characteristics using two approaches. The first one made use of Hovmöller diagrams, which display 5°-latitudinally-averaged 3-h total precipitation. The second method made use of hourly 2D-precipitation maps covering the entire analysis domain. All model data were averaged to $0.25° \times 0.25°$ to match the horizontal resolution of the TMPA precipitation product.

In the Hovmöller diagrams, westward-propagating mesoscale precipitation features were identified for daily 24-h periods, i.e., starting at 12 UTC of one day and ending at 12 UTC of the following day, to accommodate the daily re-initialization of the NWP-forecasts (Section 2.1) and ensure comparability between the different datasets. We first identified contiguous areas with 3-h rain rates greater than a given threshold $\gamma_{p,hov}$. Next, we identified the longitudinal location of the precipitation maximum in the earliest and latest 3-h window during a 24-h period of each identified feature, $\phi_1$ and $\phi_2$, respectively. We defined the so-characterized contiguous precipitation areas as representing westward-propagating features if $\phi_2 < \phi_1$ and if their duration was $\geq$6 h. $\phi_1$ and $\phi_2$ were further considered as the start and end location of the feature and were used to derive the propagation speed. We discuss the results for $\gamma_{p,hov} = 20$ mm day$^{-1}$ in detail. Choosing higher thresholds ($\gamma_{p,hov} = 25$ or 30 mm day$^{-1}$) resulted in fewer identified features, but had no effect on overall statistics. This analysis allowed estimating propagation speed, spatial extent, and precipitation rate per the identified mesoscale precipitation feature, quantities we use to evaluate the performance of the models compared to observations in Section 3.1.

We also objectively identified MCSs in the model and observational data using the hourly (3-h for observations) 2D-precipitation fields. An MCS is defined as any feature consisting of 20 or more eight-way connected $0.25° \times 0.25°$ grid-points (every grid-point that touches another grid-points' edges or corners) with a rain rate exceeding a given threshold, $\gamma_{p,2D}$, corresponding to a minimum feature size of roughly 12,500 km$^2$. We used thresholds of $\gamma_{p,2D} = 60, 70, 80,$ or 90 mm day$^{-1}$ and discussed results for $\gamma_{p,2D} = 70$ mm day$^{-1}$ in detail. We associated the center of mass of an identified object as its location at the given output time step. Note that identifying precipitation features in the Hovmöller diagrams required lower thresholds due to the applied 5°-latitudinal averaging (see above).

## 3. Results

### 3.1. Westward-Propagating Mesoscale Precipitation Features

In order to assess the impact of MCSs on the large-scale circulation and its dependency on the model used, we first needed to compare the representation in the three models used here, even though previous studies drew similar conclusions. Our results were consistent with previous studies, i.e., using a model relying on the explicit representation of convection significantly improved the representation of MCSs. We therefore keep the comparison short here.

We first assessed the ability of the models to represent the statistics of westward-propagating organized precipitation over Western Africa by applying the identification algorithm detailed in Section 2.2 to Hovmöller diagrams of three-hourly mean precipitation over West Africa. In Figure 1, we show such diagrams for 15 days of the analysis period (8 August 2016 00 UTC–22 August 2016 00 UTC, 10° N–15° N) for display purposes. The diagrams were similar for the remaining time periods of the analysis period and other latitude bands. It is evident from just eyeballing and

comparing to the observations that ICON-HiRes better captured the characteristics of observed organized westward-propagating precipitation than the other two models, IFS and ICON-oper (Figure 1). The observations display the expected pattern of mostly westward-propagating, long-lived (>24 h) precipitation features with high precipitation rates (cf. [37–42]). Non- or slowly-propagating precipitation features in the observations were associated with lower precipitation rates. As expected from other studies [15,24,27], the Hovmöller diagram derived from the ICON-HiRes simulation was also dominated by intense westward-propagating precipitation features, albeit slightly more numerous than in observations. As these simulations were re-initialized every 24 h (see Section 2.1), mesoscale precipitation features could not persist beyond the 36-h simulation cycle.

In contrast to the observations and the ICON-HiRes simulation, the corresponding Hovmöller diagrams of ICON-oper and IFS did not indicate an adequate simulation of westward-propagating precipitation. The diagrams displayed a much more smeared-out and weaker-intensity precipitation patterns compared to TMPA and ICON-HiRes. Another failure was the simulation of eastward-moving or stationary precipitation systems. The work in [25] found similar behavior in a convection-parameterizing model setup over the same region.

During August 2016, westward-propagating mesoscale precipitation features contributed 47% to the total precipitation in the TMPA data for the 10° N–15° N latitude band analyzed here. For ICON-HiRes, this contribution amounted to 65%. These values were smaller than those reported in other studies (e.g., [40,43]), but should be considered as low estimates because some evidently westward-propagating systems were missed by the identification algorithm (cf. Figure 1). In IFS and ICON-oper, the contribution amounted to only 24% and 30%, respectively.

We continue our assessment of model-simulated westward-propagating mesoscale precipitation features by showing Probability Distribution Functions (PDFs) of propagation speed, sizes, and mean rain rates in Figure 2 for the whole of August 2016. The PDFs were very similar if only features having lifetimes exceeding 12 h were considered. Overall, the agreement between TMPA and ICON-HiRes was remarkable in terms of propagation speed, object size, and precipitation intensities per feature. In contrast, ICON-oper and IFS were not able to capture the observed statistics and simulated too wide, too slow, and too weakly-precipitating objects.

Considering the evidence presented so far, we concluded that ICON-HiRes simulations, in contrast to IFS and ICON-oper, were capable of representing the main characteristics of westward-propagating precipitation systems over West Africa compared to observations. In contrast, most systems did not propagate in ICON-oper and IFS. Despite the lack of propagating features, ICON-oper and IFS were able to reproduce the mean precipitation distribution (Figure 3). In this respect, there was no clear advantage of the convection-permitting resolution. In the next section, we investigate whether the differences in the MCSs representation projected on the large-scale circulation of the atmosphere.

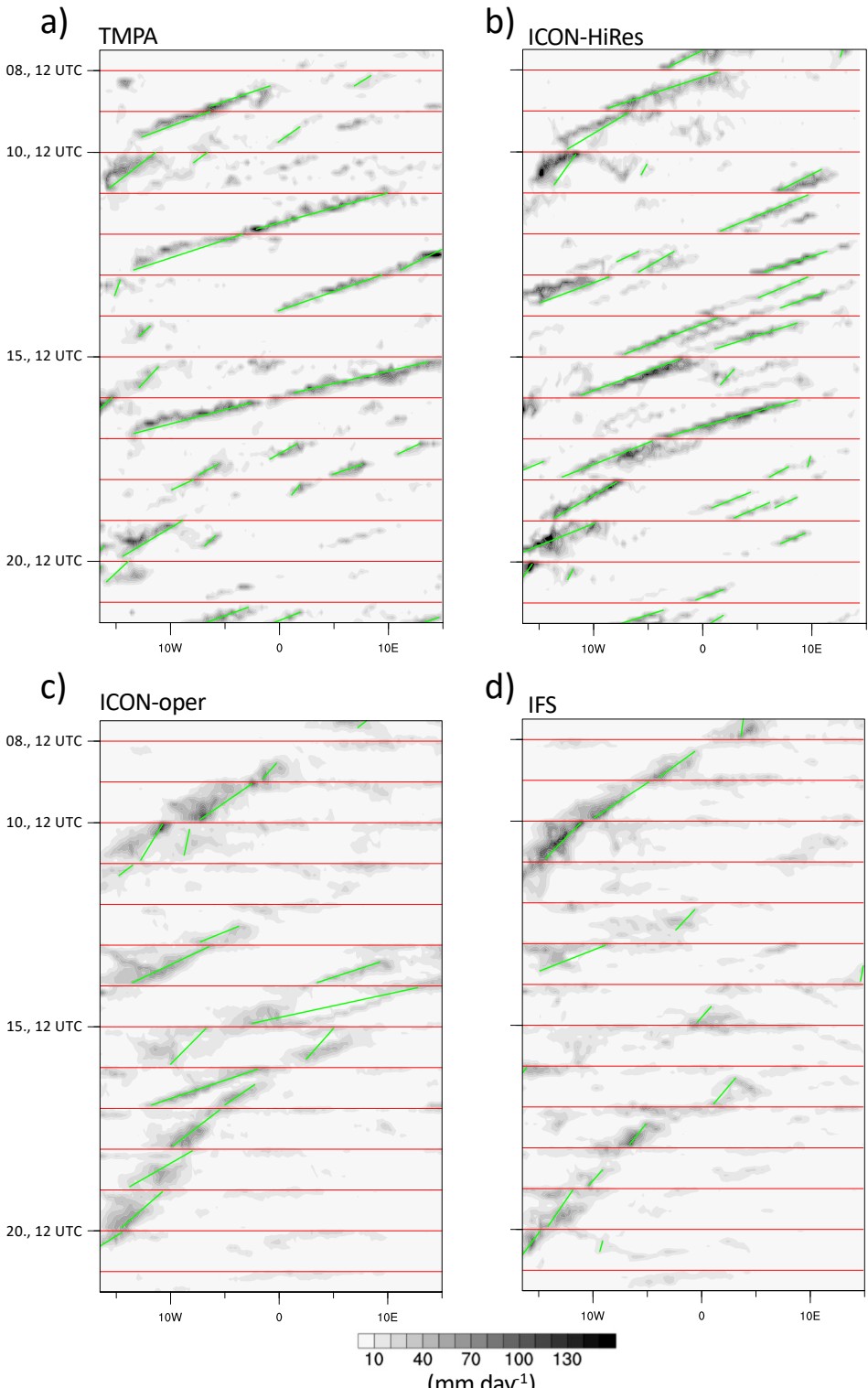

**Figure 1.** Hovmöller diagrams of three-hourly mean precipitation for the period 8 August 2016 00 UTC–22 August 2016 00 UTC over West Africa (10° N–15° N, 16.5° W–15° E, land only): (**a**) TRMM Multi-satellite Precipitation Analysis (TMPA); (**b**) the Icosahedral Non-hydrostatic model (ICON)-HiRes; (**c**) ICON-operational (oper); and (**d**) Integrated Forecast System (IFS). All model data were coarse-grained to 0.25° × 0.25°, i.e., the resolution of the observations. Green lines indicate objectively-identified westward-propagating precipitation features (see the main text for details). Horizontal lines indicate 12 UTC of every day.

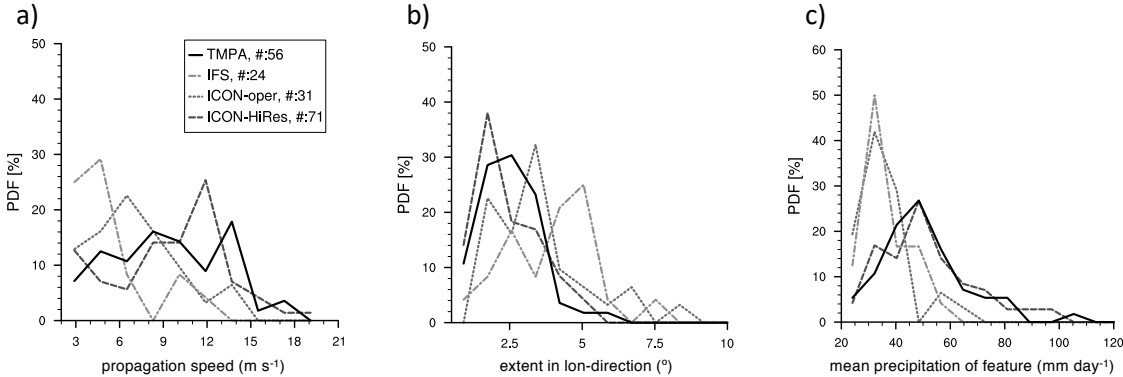

**Figure 2.** PDFs of objectively-identified westward-propagating precipitation feature characteristics (see the main text for details) (land points in 10° N–15° N, 16.5° W–15° E, 1–31 August 2016). (**a**) Propagation speed (ms$^{-1}$); (**b**) feature extent in the longitudinal direction (°-longitude); and (**c**) mean precipitation (mm day$^{-1}$). The numbers in the legends indicate the number of detected features in each dataset.

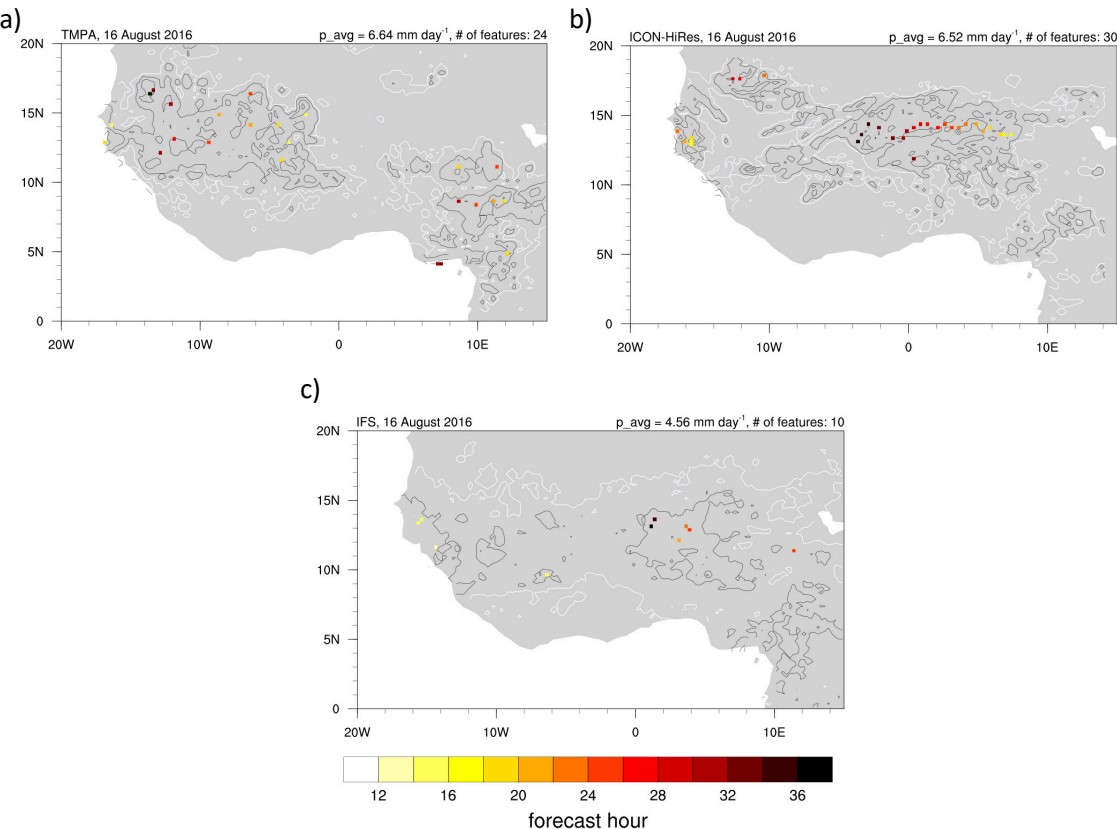

**Figure 3.** Objectively-identified precipitation features of 20 or more connected grid points with hourly rain rates exceeding a threshold of $\gamma_{p,2D}$ >70 mm day$^{-1}$ sorted by forecast hour during which they were simulated (colors) and daily surface precipitation over land (contours) for 16 August 2016: (**a**) TMPA; (**b**) ICON-HiRes; and (**c**) IFS. The contour lines denote precipitation of 1, 10, 40, 80, and 120 mm day$^{-1}$. The 1-mm day$^{-1}$ contour is shown in white. The average precipitation is shown at the top of each panel.

### 3.2. Representation of MCSs' Impacts on Continental-Scale Forecast Evolution

In the following, we used data spanning the entire simulation period (August 2016) to characterize the evolution of the forecast in ICON-HiRes and IFS depending on the presence of simulated MCSs in ICON-HiRes. We chose to compare ICON-HiRes to IFS because both models started from the

same initial conditions and the ICON-HiRes boundary conditions were provided by IFS (Section 2.1). Hence, any difference between the two models can be more clearly attributed to the representation of convection.

### 3.2.1. Identification of MCSs in 2D Precipitation Fields

Analysis of the effect of MCSs on the large-scale dynamics requires stepping away from the meridionally-averaged picture presented in the previous section. Instead, we now focus our analysis on features objectively identified in the 2D surface precipitation field (cf. Section 2.2). We illustrate our identification approach in this short section and also use it to contrast typical differences in the representation of precipitation between the models.

We show an example of our feature-identification methodology, with $\gamma_{p,2D} = 70$ mm day$^{-1}$, for data stemming from the forecasts initialized on 16 August 2016 at 00 UTC in Figure 3. Note that a single precipitation feature may contribute up to 24 (eight) identifications for a single day in the models (observations) due to the sampling of all available time steps (one-hourly for the models and three-hourly for the observations). The observations (Figure 3a) indicated zonally-oriented regions of high surface rain rates with 24 objectively-identified mesoscale precipitation objects, which mostly indicated westward propagation. ICON-HiRes precipitation (Figure 3b) showed a more zonal orientation than that in IFS (Figure 3c). Furthermore, the precipitation maxima as simulated by ICON-HiRes appeared isolated, with large areas of no precipitation between them. IFS on the other hand simulated some amount of precipitation almost everywhere in the domain considered here during the day, yet another manifestation of the well-known intermittency problem in models relying on parameterized convection (e.g., [44]). The corresponding plot for ICON-oper displayed a similar picture as for IFS (not shown). Interestingly, ICON-HiRes and IFS had about the same amount of rain (Figure 3b,c), implying that differences in the large-scale circulation between the models were not due to significantly different total precipitation.

### 3.2.2. Impacts on Forecast Evolution

We conjectured that there exists a relationship between differences in the evolutions of forecasted wind fields and the number of identified MCSs between two forecasts during one 24-h forecast. We characterized the different forecast evolutions between ICON-HiRes and IFS at every 1-h output time step by the spatially-averaged Root Mean Squared Difference (RMSD) between the forecasted total wind speeds at different heights. We considered wind speeds at 850 and 650 hPa, as these heights are of relevance for the African Easterly Jet and the African Monsoon, respectively.

An example of the evolution of the RMSD of total wind speeds at 650 and 850 hPa (10° N–15° N) over the forecast cycles of two days (9 and 16 August 2016) is given in Figure 4. In the forecasts initialized on 16 August 2016, 00 UTC (dashed lines in Figure 4), the MCS-identification algorithm detected 30 features in ICON-HiRes and just 10 in IFS ($\gamma_{p,2D} = 70$ mm day$^{-1}$). In ICON-HiRes, a series of small precipitation clusters formed at about 15 h of forecast lead time in the northeastern part of the domain and organized into a large squall line structure propagating westward throughout the remainder of the forecast. No mentionable mesoscale precipitation structures were simulated by IFS at any point during this forecast. The RMSDs of both wind fields increased over the entire duration of the forecasts (Figure 4). The corresponding spatial structures of the simulated precipitation and wind fields at 850 hPa at two forecast snapshots (17 and 30 h of forecast lead time) are shown in Figure 5e–h. The impact of the MCS simulated in ICON-HiRes on the large-scale circulation was evident and explained the increase in RMSDs over time (Figure 4).

In the forecast initialized on 9 August 2016, 00 UTC (solid lines in Figure 4), both models simulated a major mesoscale westward-propagating precipitation feature associated with an African Easterly Wave-trough located slightly eastward of the precipitation feature in both models (see also Figure 1). The corresponding spatial structures of the simulated precipitation and wind fields at 850 hPa at two forecast snapshots (20 and 30 h forecast lead time) are shown in Figure 5a–d. While

the precipitation feature took the form of an intense squall-line in ICON-HiRes (Figure 5a), IFS simulated weaker-intensity precipitation spread-out over a larger area (Figure 5b,d). The numbers of objectively-identified mesoscale precipitation features with $\gamma_{p,2D} = 70$ mm day$^{-1}$ are 39 (ICON-HiRes) and 29 (IFS) on this day. The evolution of the overall large-scale wind field remained very similar between both models, most probably due to the strong synoptic forcing imposed by the AEW trough (not shown). This was confirmed by the evolution of the RMSDs (see Figure 4, solid lines).

Framing the above in a broader context, our findings were in line with previous studies that showed that precipitation appeared rather deterministic when the large-scale forcing was strong and appeared more random if the forcing was weak [45–47]. Under weakly-forced conditions, mesoscale precipitation clusters in IFS and ICON-oper appeared at various locations in the domain around local noon and then diminished after a few hours (not shown). In ICON-HiRes, precipitation clusters forming under weakly-forced conditions, e.g., on 16 August 2016 (see above), were able to effectively organize into large and long-lived MCSs, which were then able to impact the large-scale circulation, which possibly impacted the forecast skill in a positive manner. A plethora of such cases can be found in the ICON-HiRes simulations.

To capture the impact of MCSs on forecast evolution, a more general approach is required. To do so, we characterized the evolution of the RMSD, i.e., the increase in difference, during the course of each available 24-h forecast.

We thus define a simple metric:

$$\sigma_{1,2}(x) = \max(\text{RMSD}(x)) - \min(\text{RMSD}(x)), \tag{1}$$

where $\min(\text{RMSD}(x))$ and $\max(\text{RMSD}(x))$ denote the minimum and maximum spatially-averaged RMSD-values for a given wind field $x$ during one 24-h forecast. $\sigma_1(x)$ compares ICON-HiRes to IFS. Following Equation (1), the evolution of the RMSDs on 9 and 16 August (Figure 4) was associated with low and high $\sigma_1$-values, respectively. We also compared the operational ICON-oper to the IFS forecasts in the same manner ($\sigma_2$).

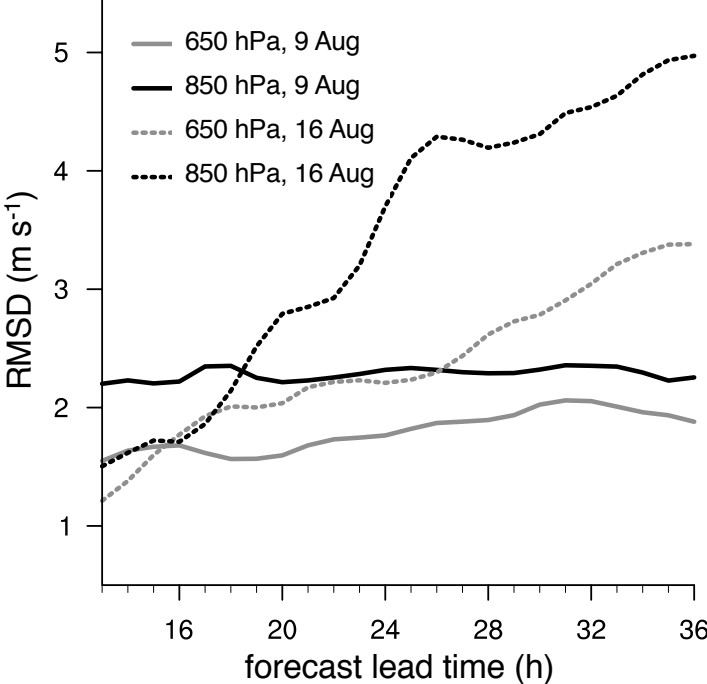

**Figure 4.** Root Mean Squared Difference (RMSD) evolution between the ICON-HiRes and IFS forecasts initialized on 9 and 16 August and for the total wind speed at 650 and at 850 hPa. Only data over land and between 10° N and 15° N are used.

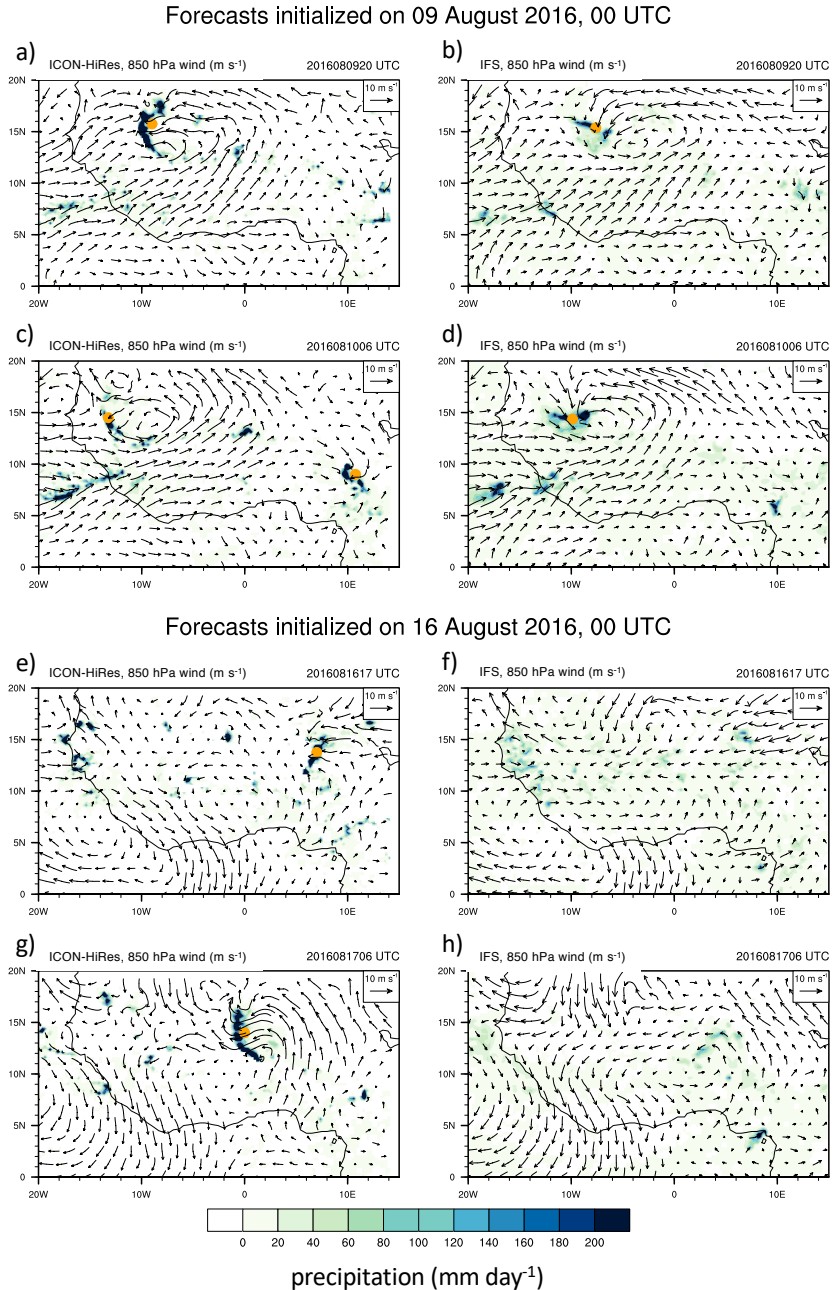

**Figure 5.** Surface precipitation and large-scale wind fields at 850 hPa from the forecasts initialized on 9 August 2016 (**a–d**) and 16 August 2016, 00 UTC (**e–h**). Data are shown for different forecast lead times: 20 (**a,b**) and 30 (**c,d**) hours for 9 August 2016; 17 (**e,f**) and 30 (**g,h**) hours for 16 August 2016 and two models: ICON-HiRes (**a,c,e,g**) and IFS (**b,d,f,g**). The wind fields represent the instantaneous values at the respective forecast lead time, and precipitation fields represent accumulations during the preceding forecast hour. Filled circles denote objectively-identified mesoscale precipitation features (cf. Section 2.2, $\gamma_{p,2D} = 70$ mm day$^{-1}$). All data are coarse-grained to $0.25° \times 0.25°$.

We show daily resolved time series of $\sigma_1$ (650 hPa) and $\sigma_1$ (850 hPa) computed for every day of August 2016 over land areas and two latitudinal bands, 5° N–10° N and 10° N–15° N, in Figure 6a,c, respectively. The difference in the numbers of objectively-identified mesoscale precipitation features with $\gamma_{p,2D} = 70$ mm day$^{-1}$ is shown in Figure 6e. The time series of $\sigma_1$ of both wind fields for the 5° N–10° N subdomain (black lines in Figure 6) showed low temporal variability, which was consistent with the generally small number of identified MCSs in both forecasts in that region (cf. Figure 6c). In contrast, considering the 10° N–15° N band, we found larger values of $\sigma_1$ with pronounced

day-to-day variability. More importantly, $\sigma_1$ tended to be smaller during periods when the difference in identified features was also small (e.g., 7–11 August, 19–21 August, 27–31 August; Figure 6c) and vice versa. This indicates that the better representation of MCSs in ICON-HiRes was able to project on the large-scale flow, even given the use of a limited area domain and of 24-h forecast. Note that the overall evolution of the time series was similar for the wind fields at both height levels, suggesting that explicitly-simulated convection had a systematic effect on the simulation of lower and middle troposphere dynamics alike.

Linear correlation coefficients between the difference in the number of identified MCSs (Figure 6c) and the time series of $\sigma_1$ (650 hPa) and $\sigma_1$ (850 hPa), $r_{\sigma_1(650)}$ and $r_{\sigma_1(850)}$, respectively, are shown in Table 1. To investigate if the signal remained visible over a larger domain, we also computed the correlation coefficients for broader latitudinal bands (9° N–17° N, 0° N–20° N, Table 1). We found notable correlations >0.5 for both wind fields, with $r_{\sigma_1(850)}$ generally being larger than $r_{\sigma_1(650)}$, indicating a stronger effect of MCSs on lower-tropospheric winds. This is consistent with the fact that propagating MCSs were associated with strong gust fronts at the surface and a shallow circulation, as will be shown in more detail in Section 4. Further, the correlation coefficients for the larger subdomains suggested an effect of MCSs on synoptic-scale dynamics. We also performed this analysis for varying values of $\gamma_{p,2D}$ (60, 80, and 90 mm day$^{-1}$) and found similar results, whereby correlations generally increased with increasing values of $\gamma_{p,2D}$.

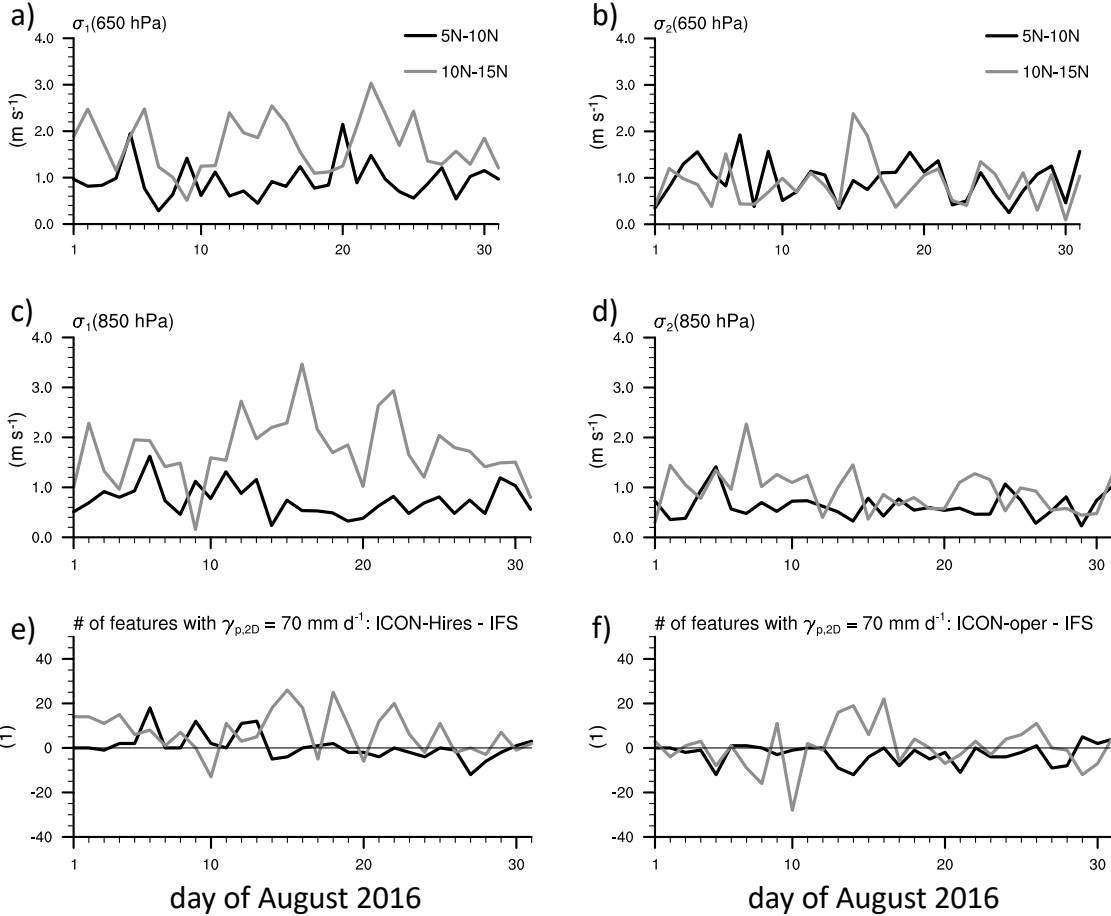

**Figure 6.** Daily resolved time series (1–31 August 2016) of $\sigma_{1,2}$ (Equation (1)) for wind fields at (**a**,**b**) 650 hPa and (**c**,**d**) 850 hPa and (**e**,**f**) the difference in the number of objectively-identified precipitation features with $\gamma_{p,2D} = 70$ mm day$^{-1}$ between IFS and ICON-HiRes (**c**) (ICON-oper (**f**)). Average values of $\sigma_1$ ($\sigma_2$) and the number of identified features are analyzed over two different subdomains and over land areas only.

**Table 1.** Linear correlation coefficients *r* between $\sigma_1$ and $\sigma_2$ for winds at the 850- and 650-hPa levels and the difference in the number of identified precipitation features with $\gamma_{p,2D} = 70$ mm day$^{-1}$ (Figure 6c) per subdomain as shown in Figure 6.

| Subdomain | $r_{\sigma_1(650)}$ | $r_{\sigma_1(850)}$ | $r_{\sigma_2(650)}$ | $r_{\sigma_2(850)}$ |
|---|---|---|---|---|
| 0° N–20° N | 0.43 | 0.53 | 0.36 | 0.19 |
| 5° N–10° N | −0.24 | 0.6 | −0.02 | −0.22 |
| 10° N–15° N | 0.41 | 0.41 | 0.2 | −0.02 |
| 9° N–17° N | 0.54 | 0.57 | 0.26 | 0.16 |

We repeated the same analysis, but comparing ICON-oper to IFS (see Figure 6b,d,f and Table 1). At first glance, the values of $\sigma_2$ generally appeared smaller than those of $\sigma_1$, and the time series indicated less temporal variability (Figure 6). However, there were some days on which the number of objectively-identified mesoscale precipitation features varied starkly between the two operational forecasts (Figure 6f). However, the linear correlation coefficients $r_{\sigma_2(650)}$ and $r_{\sigma_2(850)}$ shown in Table 1 mostly indicated no correlation. In short, this indicated that the mesoscale precipitation features in the parameterized models did not project on the mean flow. Hence, the employed convection scheme (based on [33]) seemed not to allow for notable convection–mean-flow interaction.

## 4. Discussion

The foregoing analysis indicated that it seemed to be the different representation of MCSs in ICON-HiRes compared to IFS that resulted in differences between the synoptic-scale forecasts of both modeling systems. We here briefly illustrate the most striking differences in the interaction of mesoscale precipitation features with the mean large-scale flow by presenting storm composites centered on objectively-identified precipitation maxima ($\gamma_{p,2D} = 70$ mm day$^{-1}$, Figure 7). We compiled composites from the forecasts initialized on 9 August 2016 00 UTC for ICON-HiRes, IFS, and ICON-oper. We chose this date as it was one of the few cases with a clear trackable mesoscale precipitation feature in the IFS and ICON-oper operational forecasts (see Figure 5a–d). Recall that this case was characterized by an AEW trough in the region, imposing a strong synoptic forcing on the simulated precipitation (Section 3.2.2). Both ICON-oper and IFS simulated mesoscale precipitation features at approximately the same time and location on that day. Days without such strong synoptic forcing did not result in any trackable, long-lived mesoscale precipitation features in IFS and ICON-oper (see above).

The linear precipitation front in ICON-HiRes was evidently associated with strong convergence at its leading edge and pronounced rear-inflow at 850 hPa, as well as a strengthening and weakening of the AEJ (650 hPa wind, not shown) behind and in front of the squall line (Figure 5, see also [26]). The corresponding vertical distribution (Figure 7a) of vertical wind velocity and horizontal divergence also displayed squall-line characteristics previously identified in observations and conceptual models (see [48,49] and the references therein). Although we did not explicitly calculate vertical transport of horizontal momentum, we conjectured that the simulated dynamics of the squall line system detailed above also resulted in fairly realistic redistribution of horizontal momentum as detailed in e.g., [50] or [51], an essential model capability in order to represent the MCS–mean flow interaction adequately.

Although not shown here, an impact on the horizontal wind fields at 850 and 650 hPa was virtually absent in ICON-oper and IFS compared to ICON-HiRes, which also explained the lack of correlation between $\sigma_2$ and the number of identified mesoscale precipitation features (Figure 6). In particular, the corresponding zonal cross-sections both showed ascending motion basically throughout the entire troposphere, with maximum ascent occurring in the 400–200-hPa region close to the identified storm center (Figure 7b,c). The region of ascending motion appeared zonally spread-out. The horizontal divergence pattern was also much weaker than in ICON-HiRes with little zonal variations. Similar differences in the vertical structure of MCSs were reported by [52]. All these differences made clear why the MCSs in ICON-HiRes, beyond their sheer larger number, led to a significantly different

large-scale circulation compared to the models with parameterized convection. Note that the spatial scales influenced by the MCSs went well beyond those of the systems themselves.

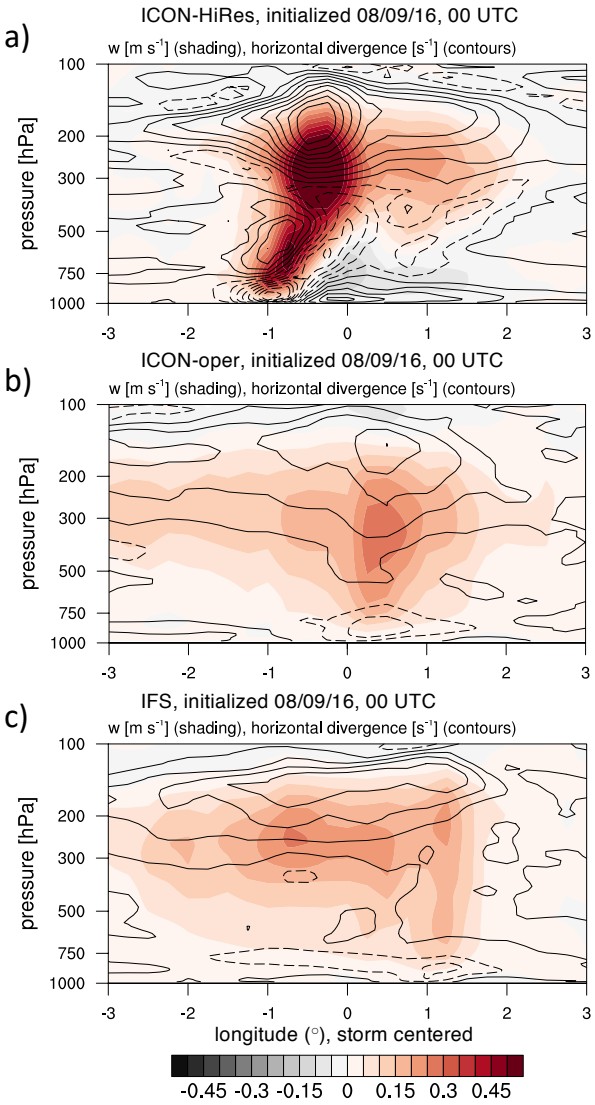

**Figure 7.** Storm-centered zonal cross-sections of vertical velocity (m s$^{-1}$) (shading) and horizontal divergence (s$^{-1}$) (contour lines; positive thin solid, zero line thick solid, negative dashed; regular spacing of $2 \times 10^{-5}$ s$^{-1}$): (**a**) ICON-HiRes; (**b**) ICON-oper; and (**c**) IFS. The fields are 2° meridionally averaged.

Our results show that performing NWP at storm-resolving model resolution over Western Africa provided a positive effect on the representation of propagating MCSs, thus improving continental-scale forecasts on timescales of 1–2 days. From our results alone, we may only speculate about positive effects on longer time and larger spatial scales. However, the work in [53], using the global ICON NWP setup with two-way nesting to a higher-resolved, convection-permitting domain over Western Africa, very recently showed that representing Sahelian storms had positive impacts on weather forecast over remote regions, e.g., mid-latitude Europe. We here provide an analysis of the mechanisms leading to these positive effects: Not only the representation of propagating MCSs was in substantially better agreement with observations, but the explicit treatment of convection allowed fully representing the complex interaction of MCS dynamics with the large-scale flow field and projects on the large scale even on a daily timescale. Whilst the ameliorated statistics of MCSs was well known from past studies, the effect of a biased representation of MCSs on the large-scale circulation had not been

quantified in such detail before. To account for MCS–mean-flow interactions fully, such forecasts should be conducted over domain sizes and time periods allowing one to encompass the full spectrum of system sizes, downstream effects on the mean flow, and lifetimes, i.e., at least a 3000-km domain edge length and 72-h forecast lead time. Two-way nesting can also be a very promising approach (see [53]). Furthermore, climate simulations could also benefit from convection-permitting modeling frameworks because the biased representation of MCSs in models relying on parameterized convection can potentially affect the climatology of the large-scale circulation.

## 5. Summary

We analyzed a suite of one observational dataset and three different model simulations over Western Africa, spanning the time period from 1–31 August 2016, to evaluate the potential effect of a misrepresentation of mesoscale convective systems (MCSs) on the forecasted 650-hPa and 850-hPa wind fields on daily timescales over a large domain. These two heights are important for characterizing the African Easterly Jet (AEJ) and West African Monsoon flows, respectively.

To do so, we used convection-permitting simulations with a grid spacing of 2.5 km that were conducted in the context of the NARVAL II field campaign (ICON-HiRes [12]; see [16] for an evaluation of bulk quantities). The simulations were daily forecasts, spanning the time period 1–31 August 2016, and covered the entire domain of the Tropical Atlantic [12]. We compared these simulations against the operational global weather forecasts from ECMWF (IFS) and DWD (ICON-oper). Both of those models relied on parameterized convection, and the analyzed data corresponded to the same experimental setup as ICON-HiRes (daily re-initialization, analysis of a 36-h forecast whilst omitting the first 12 h for spin-up reasons). We also compared the three model simulations against an observational dataset for reference (TMPA, [35,36]). In contrast to previous studies mainly focusing on the analysis of precipitation and MCS statistics, we focused our analysis on the bias in the 850- and 650-hPa large-scale wind fields arising on daily timescales due to the biased representation of MCSs in global NWP models relying on parameterized convection.

We objectively identified mesoscale precipitation features in all four datasets and found that for IFS and ICON-oper, the identified features did not compare well to the observations, which showed the clear and well-known signature of westward-propagating MCSs. In contrast, ICON-HiRes matched the observed propagation speed, size, and mean rain rate of mesoscale precipitation features remarkably well, although their occurrence frequency was overestimated.

We analyzed the difference in the large-scale wind fields at 850 and 650 hPa between daily forecasts of ICON-HiRes and IFS and found that to first order, the difference between ICON-HiRes and IFS scaled with the difference in the simulated number of propagating mesoscale precipitation features. The higher the difference, the larger the difference in the simulated wind fields, both at 850 and 650 hPa. This was the case because in ICON-HiRes, the mesoscale precipitation features mostly took the form of intense, long-lived squall lines with strong dynamical features. This was particularly visible on days with weak synoptic forcing, i.e., on days not characterized by the presence of an African Easterly Wave in the region. In contrast, the composite precipitation fields appeared wide-spread and smeared-out, and the horizontal large-scale wind fields at 850 and 650 hPa showed no discernible response to the deep convection in IFS and ICON-oper, despite the relatively high spatial resolution of these global NWP models. Albeit covering similar spatial scales, the magnitudes of the storm-composited vertical motion and divergence were about half of those simulated by ICON-HiRes and did not even closely match the complexity of those simulated in ICON-HiRes. In fact, the large-scale flow appeared to dominate the mesoscale precipitation distribution in IFS an ICON-oper. This was contrary to the behavior in ICON-HiRes, in which the MCSs induced their own mesoscale circulation feeding back to the large-scale flow.

To conclude, although the two operational state-of-the-art (and heavily tuned) global NWP models we analyzed here provided skillful estimates of West African surface rainfall, they showed a biased representation of mesoscale precipitating systems displaying notable interaction with the mean flow,

an aspect column physics fails to provide by definition. This feed back on the large-scale circulation, especially in synoptic situations not dominated by prominent large-scale forcing, e.g., an African Easterly Wave passing through the domain. In fact, it has been recently demonstrated that this feedback can improve operational weather forecasts in the mid-latitudes [53]. This speaks for the use of convection-permitting simulations on very large domains. However, we must also not forget the required computational resources going along with such endeavors and that global models relying heavily on parameterized deep convection will continue being a staple of everyday atmospheric science [7]. In that context, simulations such as ICON-HiRes will play an invaluable role in shaping our conceptual understanding of atmospheric convection and in the development of urgently-needed next-generation parameterization schemes thereof [3,54].

**Author Contributions:** Conceptualization, K.P., C.H., and D.K.; formal analysis, K.P.; funding acquisition, C.H.; investigation, K.P.; methodology, K.P., C.H. and D.K.; resources, D.K.; supervision, C.H. and D.K.; visualization, K.P.; writing, original draft, K.P.; writing, review and editing, K.P., C.H., and D.K.

**Funding:** This research was carried out in the Hans Ertel Center for Weather Research (HErZ). This German research network of universities, research institutions, and the German Weather Service (DWD) is funded by the BMVI(Federal Ministry of Transport and Digital Infrastructure).

**Acknowledgments:** We are grateful to Jürgen Bader (MPI-M), two anonymous reviewers, and the Academic Editor of this Special Issue for useful comments on the manuscript. The ICON-HiRes simulations were conducted on the supercomputer system of the European Centre for Medium-Range Weather Forecasts (ECMWF), and the simulation data are stored at the the German Climate Computing Center (DKRZ). We acknowledge ECMWF and DWD for providing the operational weather forecast data used in this study.

**Conflicts of Interest:** The authors declare no conflict of interest. The funders had no role in the design of the study; in the collection, analyses, or interpretation of data; in the writing of the manuscript; nor in the decision to publish the results.

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

**Sample Availability:** The underlying primary data and analysis scripts are available upon contact to the first author and will be made publicly and openly available through the long-term archiving service at DKRZ.

