# Peer review of "Different Representation of Mesoscale Convective Systems in Convection-Permitting and Convection-Parameterizing NWP Models and Its Implications for Large-Scale Forecast Evolution"

_atmosphere, doi:10.3390/atmos10090503_

Round 1

Reviewer 1 Report

This manuscript addresses an important issue: the potential effects of a biased representation of mesoscale convective systems in numerical models relying on parameterized convection. The manuscript is clearly written and well structured. I recommend its acceptance for publication after a minor revision.

Line 213: Add "on" after "impacts". Line 271: Delete "again". Figure 5: Consider adding one column of plots to show differential wind fields (i.e., ICON-HiRes minus IFS) for a more clear demonstration of the impact of MCSs on large-scale circulation. Figure 7: A similar structural difference between convection-permitting and convection-parameterizing simulations was reported in a very relevant study by Moncrieff and Liu (2006).Moncrieff, M. W, and C.-H. Liu, 2006: Representing convective organization in prediction models by a hybrid strategy. J. Atmos.Sci., 63, 3404-3420. Line 406: Concluding -> To conclude.

Author Response

Please see the attached PDF file.

Reviewer 2 Report

The authors present work considering the impact of Mesoscale Convective Systems (MCSs) on the large-scale flow in Western Africa. They compare simulations made in 2 convection-parameterized models against a convection-permitting model and observations. They show that the MCSs influence the flow at 850 and 650 hPa in the convection-permitting but not in the convection-parameterizing. This study shows merit, however I have a couple comments which I feel will make the arguments presented stronger.

General:

I am a little confused as to why two different forecasts/MCSs are compared for the composites: 16 August for HiRes and 9 August for the other models/resolutions. Surely it would be fairer and more supportive of your arguments if you created composites for all of the models on the same day (for example, 9 August as you have a tracked MCS in that one)? Currently, it could be argued that there are differences because you are looking at MCSs on different days rather than because there are differences in the way the MCSs are being simulated.

I also wonder if the conclusions from Figure 5 might be more strongly supported by showing just one of the levels (for example, 850 hPa) for the current forecast used (16 August) and adding in the ones at 850 hPa for the forecast initiated on 9 August.

Specific:

Please be consistent with spelling of parameterization/parameterized - some have been spelled with an 's', others with a 'z'.

line 8: In the abstract it would be good to state how the representation is remarkable: is it remarkable in a good way or a bad way? Is it remarkable because there are more or because there are less than expected?

line 20: change to ''puts the largest''

line 21: 'change to 'When asked with''

line 139: It would be useful to indicate here the reason for putting everything onto a 0.25x0.25 grid here. I know this is mentioned in a figure caption, but I feel this should be mentioned in the main text.

lines 209-212: The text here is a little unclear.

Figure 4: Dates do not match up. The caption, main text and legend are for 9 and 16 August but the title says RMSDs for 16 & 22 August - please correct.

Line 277 (and in abstract): I'm not sure about the use of the word spread - in this context I would prefer differences as you are not discussing the spread of an ensemble, you are discussing the differences between two forecasts.

Figure captions of figures 6 and 7: remove "See main text for details".

Author Response

Please see the attached PDF file.

Round 2

Reviewer 2 Report

Thank you for revising the paper. I am glad that you found my comments helpful. I have no further comments to raise on your manuscript, well done on a great piece of work.

Author Response

Response to Review #2 of Reviewer2

Please note that any changes made to the previously submitted manuscript are marked in the resubmitted *_diff.pdf file. Deleted passages are striked through in red, additional text is marked in blue.

Thank you for revising the paper. I am glad that you found my comments helpful. I have no further comments to raise on your manuscript, well done on a great piece of work.

Thank you very much. We again sincerely thank the reviewer for the comments and issues raised in the first round of reviews which helped enhance the clarity of our manuscript.